# Genetic Diversity and Mating System of Two Mangrove Species (*Rhizophora apiculata* and *Avicennia marina*) in a Heavily Disturbed Area of China

Wenxun Lu [1,2], Zhen Zou [2], Xueying Hu [1] and Shengchang Yang [2,3,*]

1   College of Life Sciences, Peking University, Beijing 100871, China; wenxunlu@pku.edu.cn (W.L.); huxueying@pku.edu.cn (X.H.)
2   College of the Environment and Ecology, Xiamen University, Xiamen 361102, China; zouzhen29@163.com
3   Key Laboratory of the Ministry of Education for Coastal and Wetland Ecosystems, Xiamen University, Xiamen 361102, China
*   Correspondence: scyang@xmu.edu.cn

**Abstract:** Mangrove forests are distributed in the intertidal zones of tropical and subtropical regions, and have been severely damaged by anthropogenic activities, climate change, and stochastic events. Although much progress has been made in the conservation and restoration of mangroves in China, studies of the genetic diversity of mangroves are lacking, especially for isolated populations, yet such studies are essential for guiding conservation and restoration efforts. Here, we evaluated the genetic diversity, spatial genetic structure, and mating system of two mangrove species, *Rhizophora apiculata* and *Avicennia marina*, in a heavily disturbed area in Tielu Harbor, Sanya City, Hainan Island, China, using 18 nuclear microsatellite markers. We found that the genetic diversity of *R. apiculata*, which is classified as 'Vulnerable' in the China Red List categories, was high and similar compared with the genetic diversity estimates of other populations reported in previous studies. In contrast, the genetic diversity of *A. marina*, which is classified as a species of 'Least Concern', was low compared with the genetic diversity estimates of other populations. We then evaluated the presence of genetic bottlenecks, spatial genetic structure, and the mating system to determine the effects that habitat destruction has had on these two species. Our findings indicate that distinct conservation and restoration approaches are needed for these two species. Generally, our results provide valuable information that will aid the development of conservation and restoration strategies for the mangroves of Tielu Harbor.

**Keywords:** genetic diversity; spatial genetic structure; mating system; *Rhizophora apiculata*; *Avicennia marina*; habitat destruction; conservation; restoration

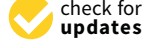



## 1. Introduction

Mangrove forests, which are dominated by plants in the Rhizophoraceae, Verbenaceae, and Combretaceae families, occur in the intertidal zones of tropical and subtropical regions [1,2]. Mangrove ecosystems are some of the world's most biodiverse and productive forest ecosystems [3], and they provide important ecosystem services, such as carbon sequestration, wave attenuation, and refuges for organisms [4,5]. Despite the high social, economic, and ecological value of mangroves, they have been severely degraded due to anthropogenic activities [6,7], climate change [8], and stochastic events [9]. For example, many mangroves have been converted to nursery areas for fishery species and play a critically important role in sustaining production in coastal fisheries [10]. Mangrove conservation and restoration strategies have been widely implemented given the alarming rate of decline in the area of mangrove forests worldwide [10–13].

Mangrove forests in China are mainly distributed along the southeastern coast, including Hainan, Guangxi, Guangdong, Fujian, and Taiwan Provinces [14]. China's mangrove

forests have been severely fragmented in previous decades, and China has historically had one of the highest rates of mangrove loss worldwide [14–16]. The area of mangrove forests in China declined from 48,300 hm$^2$ in 1950 to 22,025 hm$^2$ in 2001 [16–18]. The implementation of strict protection and large-scale restoration measures after 2001 has resulted in an increase in the area of mangrove forests; by 2019, the area of mangroves in China reached 30,000 hm$^2$ [19,20]. Despite this conservation success, an increase in the area of mangroves does not, by itself, result in the restoration of a healthy mangrove ecosystem [16], as habitat degradation, decreases in biodiversity, and biological invasions are still major problems affecting mangroves in China [21–23].

Reductions in population size are the most direct effect of anthropogenic activities and climate change on populations [24]; however, anthropogenic activities and climate change can also affect other characteristics of populations, such as genetic diversity, the mating system, and spatial genetic structure. Genetic diversity refers to the genetic variation among different individuals or populations within a species [25], and it buffers populations against variability in environmental conditions [26]. Thus, characterizing the genetic diversity of populations is a major goal of research in biodiversity conservation [27,28]. The effects of anthropogenically driven habitat fragmentation and destruction on the genetic diversity of populations of species have been well studied [29,30]. Habitat fragmentation reduces the size of populations and increases their isolation, which results in reduced heterozygosity, the loss of alleles, and reduced gene flow between populations. This can lead to an increase in inbreeding and reductions in effective population size and can have deleterious effects on the long-term persistence of populations [31,32]. Small population size and the isolation of populations can have negative effects on the mating system of plants [33]. Some species also possess ecological and genetic traits, such as those that prevent self-pollination, that counteract the deleterious effects of small population size and isolation on mating systems [34]. Spatial genetic structure is another important component of the population genetics of a species, as a thorough understanding of the spatial genetic structure within and among populations can aid in the development of conservation and restoration strategies [35,36]. An increasing number of studies have examined the population genetics of mangroves in recent years, and investigations of the population genetic diversity and spatial genetic structure of different species in mangroves have been conducted to guide mangrove conservation and restoration efforts [9,37,38].

Tielu Harbor Mangrove Nature Reserve is located in Sanya City in southern Hainan Island. A total of 13 true mangrove species in nine genera and six semi-mangrove species in six genera have been documented in the reserve [39]. Mangrove resources are abundant in Tielu Harbor, and the mangroves in this area are ancient and endangered [14]. Previous studies indicate that many ancient trees of the mangrove species, such as *Bruguiera sexangular*, *Bruguiera gymnorhiza*, *Lumnitzera racemose*, *Lumnitzera littorea*, and *Xylocarpus granatum*, occur in Tielu Harbor [14,39,40]. Increases in human activities have reduced the area of mangroves in Tielu Harbor, which is currently approximately 3–4 hm$^2$ [14,40], and degraded existing mangrove habitat. The mangroves of Tielu Harbor are thus in a precarious state and require urgent conservation attention [14,39].

Mangroves in Tielu Harbor can be divided into four major types: *Rhizophora apiculata* communities, *Avicennia marina* communities, *Lumnitzera racemosa* communities, and *Xylocarpus granatum* communities [14,39]. Of these, *R. apiculata* is the dominant species, and *A. marina* is generally considered a pioneer species (Figure 1). *R. apiculata* and *A. marina* are listed as 'Vulnerable' and of 'Least Concern', respectively, in the China Red List categories [16]. Study of the genetic diversity, mating systems, and population genetic structure of these two important mangrove species is needed to ensure that effective conservation measures are taken for the mangrove forests in Tielu Harbor.

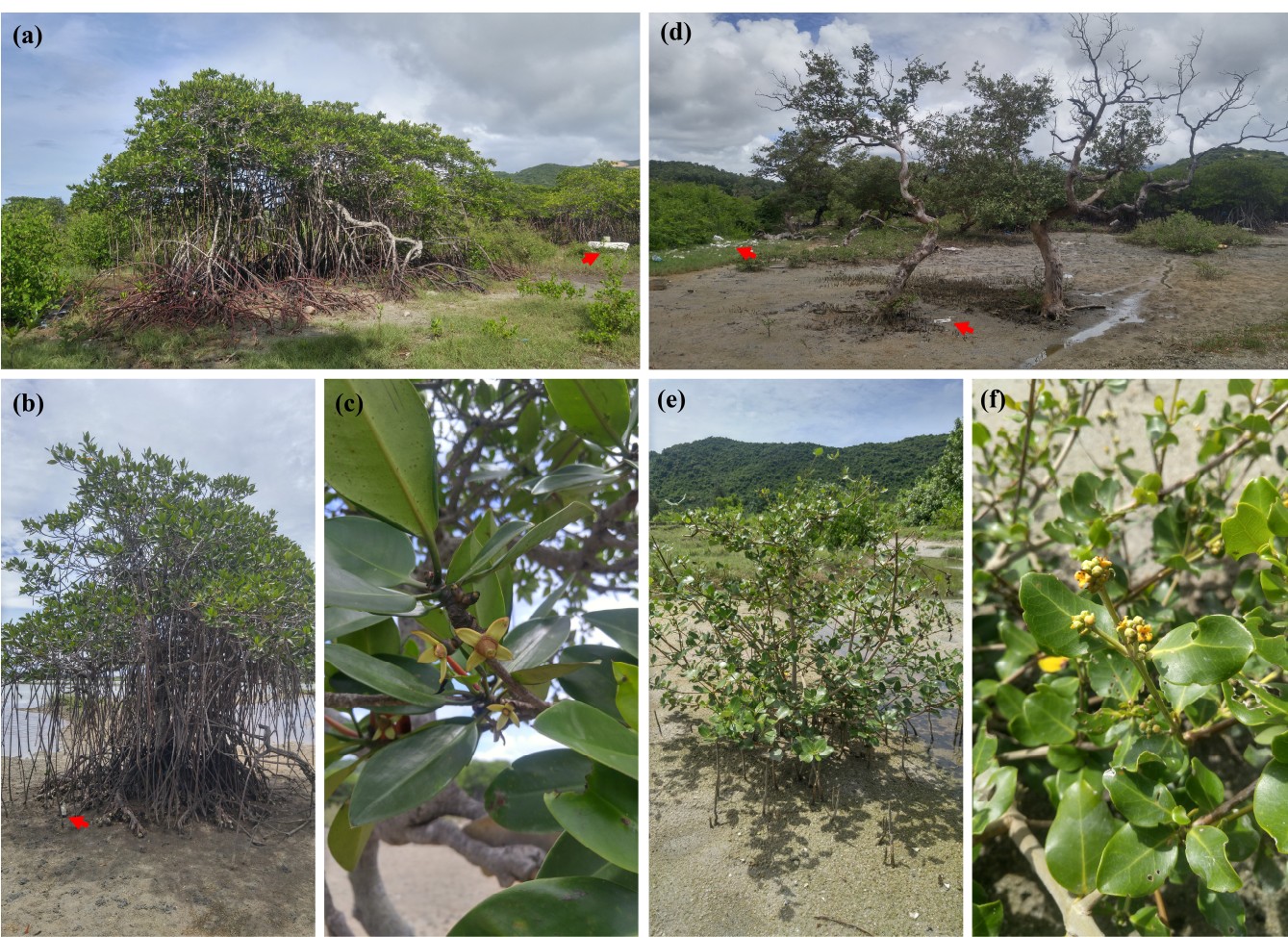

**Figure 1.** Images of *Rhizophora apiculata* and *Avicennia marina* in Tielu Harbor Mangrove Nature Reserve. Individuals of *R. apiculata* on land (**a**) and near the sea (**b**) and flowers of *R. apiculata* (**c**). Individuals of *A. marina* on land (**d**) and near the sea (**e**) and flowers of *A. marina* (**f**). The red arrows indicate pieces of garbage in the mangroves.

Here, we investigated the genetic diversity, spatial genetic structure, and mating systems of two mangrove species, *R. apiculata* and *A. marina*, in Tielu Harbor Mangrove Nature Reserve. We also conducted a comprehensive analysis of our results and recommended specific conservation measures that could be taken to guide ongoing mangrove conservation and restoration efforts in Tielu Harbor.

## 2. Materials and Methods

### 2.1. Study Area and Plant Sampling

Plant materials for this study were collected from Tielu Harbor Mangrove Nature Reserve, Sanya City, Hainan Province, China (18°15′–18°17′ N, 109°42′–109°42′ E, Figure 2). Tielu Harbor features a tropical monsoon climate, and the mean annual precipitation and temperature are 1255 mm and 25.5 °C, respectively [39]. Sampling was conducted from June to August 2016. Two quadrats were established for *R. apiculata* ($80 \times 130$ m$^2$) and *A. marina* ($110 \times 180$ m$^2$). Leaves of all adult individuals and some seedlings of the two species in quadrats were sampled. The coordinates of each sample were recorded with a Garmin GPSmap 60CSx (Garmin Ltd., Lenexa, KA, USA). A total of 167 samples (139 adults and 28 seedlings) and 210 samples (152 adults and 58 seedlings) were collected for *R. apiculata* and *A. marina*, respectively. In addition, three individuals of *R. apiculata* and five individuals of *A. marina* with propagules were randomly selected and used as mother trees

for paternity analysis. To minimize the impact on the capacity for population renewal, approximately 10 propagules were randomly sampled from each mother tree. Leaves and propagules collected were desiccated in plastic zip-lock bags with silica gel and stored at room temperature until DNA extraction.

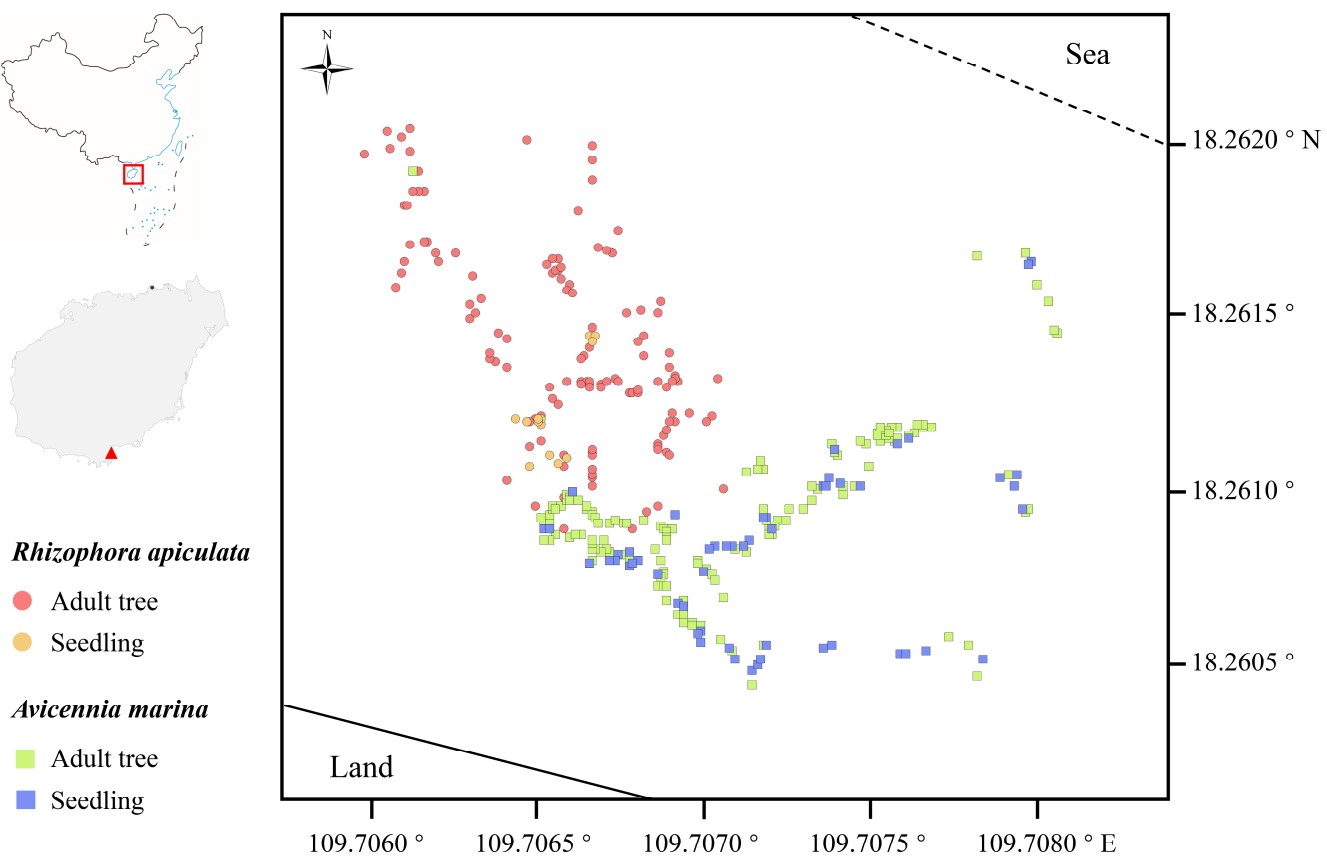

**Figure 2.** Map of *Rhizophora apiculata* (circles) and *Avicennia marina* (squares) individuals in quadrats.

*2.2. DNA Extraction, PCR Amplification, and Microsatellite Screening*

The total genomic DNA was extracted from the dried leaves and propagules of each sample using a Plant Genomic DNA Kit DP305 (Tiangen Biotech, Beijing, China). Specifically, for *A. marina*, we collected their propagules and cultured them until new leaves appeared in the laboratory, and then collected leaves of each seedling for DNA extraction. For *R. apiculate*, we directly collected a part of each propagule for subsequent DNA extraction because of its large propagule. After the concentration and purity were determined, each DNA sample was diluted with TE buffer to a concentration of 5 to 20 ng/μL and stored at −20 °C until use. Eighteen nuclear microsatellite markers (nine nuclear simple-sequence repeat (nSSR) markers for each species) developed by previous studies [41–44] were amplified for *R. apiculata* and *A. marina* (Supplementary Table S1). Fluorescently labeled primers (with HEX or ROX) were synthesized (Sangon Biotech [Shanghai] Co., Ltd.), and the PCR amplification conditions used were based on a previously published protocol [41–44]. All amplified PCR products were screened using capillary electrophoresis (Sangon Biotech Co., Ltd., Shanghai, China).

*2.3. Statistical Analysis of Genetic Parameters*

For each nuclear locus, the Hardy–Weinberg equilibrium (*HWE*), linkage disequilibrium (*LD*), and null alleles were tested using GenePop Volume 3.4 [45]. Genetic diversity parameters, such as the number of alleles (*Na*), number of effective alleles (*Ne*), Shannon diversity index (*I*), observed heterozygosity (*Ho*), expected heterozygosity (*He*), and unbiased

expected heterozygosity (*uHE*), were calculated using GenAlex Version 6.5 [46]; ADZE Version 1.0 [47] was used to determine the allele abundance (*AR*) and private allele abundance (*PAR*). The inbreeding coefficient (*Fis*) and its 95% confidence interval was calculated using GDA v1.1 (http://en.bio-soft.net/dna/gda.html, accessed on 28 December 2021). The total paternity exclusion probability of the first [Pr(*Ex1*)] and second parent [Pr(*Ex2*)] was estimated using CERVUS Version 3.0 [48]. In addition, we performed the analysis of molecular variance (AMOVA) to estimate the distribution of genetic variance among adults and seedlings using $\Phi$-statistics with GenAlEx Version 6.5 [46] for both two species.

### 2.4. Analysis of Bottlenecks and Spatial Genetic Structure

The likelihood of prior population bottlenecks in the two species was estimated using BOTTLENECK Version 1.2.02 [49] using three models, the two-phase model (TPM), the infinite allele model (IAA), and the stepwise mutation model (SMM), under default settings. The significance of bottlenecks was estimated using the sign test and one-tailed Wilcoxon sign-rank test.

Spatial autocorrelation analyses were carried out for the entire population and for seedlings and adults using SPAGeDi Version 1.2 [50]. Average multilocus kinship coefficients (*Fij*) were calculated for nine distance classes according to size of quadrats and a previous study [51]. Specifically, for *A. marina*, distance classes were 1, 2, 4, 8, 16, 32, 64, 128 and 300 m; and we set distance classes as 1, 2, 4, 8, 16, 32, 64, 128 and 200 m for *R. apiculate*. The 95% confidence interval of the different distance classes was tested using 10,000 random permutations. The *Sp* statistic was calculated for individuals from the first distance class following the method of Vekemans and Hardy [52].

### 2.5. Analysis of Mating System Parameters and Pollen Dispersal

MLTR Version 3.2 [53] was used to analyze the multilocus outcrossing rate (*tm*), the single-locus outcrossing rate (*ts*), and the biparental inbreeding rate (*tm*−*ts*) under the mixed mating model with 1000 bootstrap replications to assess the 95% confidence intervals for standard errors. The most likely pollen donor for each propagule was determined via maximum-likelihood assignment in CERVUS Version 3.0 [48]. For paternity analyses, all adult individuals were considered as candidate parents, and 10,000 simulations were conducted with 0.01 as the mistyped rate, 0.9 as the sampled candidate parent proportion, and 95% as the strict and 80% as the relaxed confidence level.

## 3. Results

### 3.1. Genetic Diversity of R. apiculata and A. marina

Two nSSR loci (RM102 and Am98) were not included in subsequent analyses because they were not polymorphic. A total of 37 alleles of eight nSSR loci and 23 alleles of eight nSSR loci were detected in both adults and seedlings of *R. apiculata* and *A. marina*, respectively. The genetic diversity parameters of each nSSR locus for all individuals of the two species are shown in Table 1. The average frequency of null alleles in *R. apiculata* and *A. marina* was 18.8% (3.5–47.1%) and 20.7% (8.4–29.3%), respectively. Only four out of 28 pairwise comparisons showed significant *LD* in the *R. apiculata* population (including seedlings, $p < 0.01$), whereas approximately half (13 out of 28) of the pairwise comparisons showed significant *LD* in the *A. marina* population (including seedlings, $p < 0.01$). Six nSSR loci (Rhst01, Rhst11, Rhst13, Rhst20, RM114, and RM116) deviated significantly from *HWE* in the *R. apiculata* population (heterozygote deficiency, $p < 0.01$), and all nSSR loci in the *A. marina* population showed heterozygote deficiency ($p < 0.01$).

**Table 1.** Genetic diversity of the selected nSSR primer pairs in all individuals of *R. apiculata* and *A. marina*.

| Locus | *n* | *Na* | *Ne* | *Ho* | *He* | *uHe* | *I* | *Pr(Ex1)* | *Pr(Ex2)* |
|---|---|---|---|---|---|---|---|---|---|
| *R. apiculata* | | | | | | | | | |
| Rhst01 | 167 | 2 | 1.519 | 0.018 | 0.342 | 0.343 | 0.525 | 0.962 | 0.880 |
| Rhst02 | 167 | 4 | 1.877 | 0.665 | 0.467 | 0.469 | 0.747 | 0.871 | 0.755 |
| Rhst11 | 167 | 2 | 1.916 | 0.347 | 0.478 | 0.479 | 0.671 | 0.903 | 0.828 |
| Rhst13 | 167 | 7 | 2.065 | 0.413 | 0.516 | 0.517 | 0.920 | 0.861 | 0.743 |
| Rhst20 | 167 | 8 | 4.322 | 0.311 | 0.769 | 0.771 | 1.589 | 0.630 | 0.452 |
| RM113 | 167 | 5 | 2.453 | 0.593 | 0.592 | 0.594 | 1.056 | 0.814 | 0.671 |
| RM114 | 167 | 6 | 2.207 | 0.467 | 0.547 | 0.548 | 0.999 | 0.839 | 0.692 |
| RM116 | 167 | 3 | 1.068 | 0.042 | 0.064 | 0.064 | 0.155 | 0.997 | 0.960 |
| Mean | 167 | 4.6 | 2.178 | 0.357 | 0.472 | 0.473 | 0.833 | / | / |
| *A. marina* | | | | | | | | | |
| Avma01 | 210 | 3 | 1.840 | 0.262 | 0.457 | 0.458 | 0.662 | 0.881 | 0.813 |
| Avma02 | 210 | 5 | 1.479 | 0.248 | 0.324 | 0.325 | 0.596 | 0.930 | 0.823 |
| Avma16 | 210 | 4 | 1.975 | 0.238 | 0.494 | 0.495 | 0.722 | 0.874 | 0.804 |
| Avma17 | 210 | 5 | 2.795 | 0.476 | 0.642 | 0.644 | 1.201 | 0.797 | 0.634 |
| Am3 | 210 | 2 | 1.379 | 0.157 | 0.275 | 0.275 | 0.447 | 0.975 | 0.901 |
| Am32 | 210 | 3 | 1.100 | 0.010 | 0.091 | 0.091 | 0.212 | 0.999 | 0.976 |
| Am40 | 210 | 4 | 1.776 | 0.152 | 0.437 | 0.438 | 0.671 | 0.903 | 0.819 |
| Am81 | 210 | 4 | 1.547 | 0.014 | 0.354 | 0.354 | 0.594 | 0.908 | 0.818 |
| Mean | 210 | 3.5 | 1.750 | 0.197 | 0.387 | 0.389 | 0636 | / | / |

**Note**: Genetic diversity parameters were estimated for combined adult and seedling plants; **Abbreviations**: *n*, number of individuals; *Na*, number of alleles; *Ne*, effective number of alleles; *Ho*, observed heterozygosity; *He*, expected heterozygosity; *uHe*, unbiased expected heterozygosity; *I*, Shannon's index of diversity; **[Pr(Ex1)]** and **[Pr(Ex2)]**, exclusion probability of the first and second parent, respectively.

The results of the genetic diversity analysis of different age groups (adults and seedlings) of the two species are shown in Table 2. Because the number of seedlings sampled was small, several genetic diversity parameters (*Na*, *I*) were lower in seedlings than in adults in the *R. apiculata* population. A similar pattern was observed in the *A. marina* population. Because *Na* is sensitive to sample size, we used ADZE Version 1.0 to calculate allele richness. In the *R. apiculata* population, the *AR* of seedlings and adults was similar, but the *PAR* of seedlings was lower than that of adults. The *AR* and *PAR* of seedlings were higher than that of adults in the *A. marina* population (Table 2). AMOVA analysis for adults and seedlings of *R. apiculata* revealed that the genetic differences mainly existed among different individuals (99.00%), only 1.00% occurred between different age groups (Supplementary Table S2). Similarly, AMOVA analysis in *A. marina* population revealed that 100% genetic variation existed within different individuals (Supplementary Table S3).

**Table 2.** Genetic diversity parameters for adults and seedlings of *R. apiculata* and *A. marina*.

| Species | *n* | *Na* | *Ne* | *Ho* | *He* | *uHe* | *I* | *Fis* | *AR* | *PAR* |
|---|---|---|---|---|---|---|---|---|---|---|
| *R. apiculata* | | | | | | | | | | |
| Adults | 139 | 4.625 | 2.165 | 0.366 | 0.464 | 0.466 | 0.828 | 0.215 | 3.046 | 0.433 |
| | | (0.800) | (0.365) | (0.084) | (0.073) | (0.074) | (0.152) | / | (0.441) | (0.130) |
| Seedlings | 28 | 3.375 | 2.169 | 0.313 | 0.486 | 0.495 | 0.813 | 0.373 * | 2.941 | 0.327 |
| | | (0.460) | (0.262) | (0.081) | (0.071) | (0.072) | (0.135) | / | (0.402) | (0.155) |
| Total | 167 | 4.625 | 2.178 | 0.357 | 0.472 | 0.473 | 0.833 | 0.246 | / | / |
| | | (0.800) | (0.341) | (0.083) | (0.072) | (0.073) | (0.149) | / | / | / |

**Table 2.** *Cont.*

| Species | n | Na | Ne | Ho | He | uHe | I | Fis | AR | PAR |
|---|---|---|---|---|---|---|---|---|---|---|
| *A. marina* | | | | | | | | | | |
| Adults | 152 | 3.500 | 1.750 | 0.197 | 0.387 | 0.389 | 0.636 | 0.497 * | 2.623 | 0.277 |
| | | (0.267) | (0.181) | (0.056) | (0.060) | (0.061) | (0.102) | / | (0.233) | (0.111) |
| Seedlings | 58 | 2.875 | 1.699 | 0.190 | 0.372 | 0.376 | 0.625 | 0.497 * | 2.704 | 0.358 |
| | | (0.389) | (0.181) | (0.049) | (0.055) | (0.055) | (0.097) | / | (0.329) | (0.143) |
| Total | 210 | 3.750 | 1.736 | 0.195 | 0.384 | 0.385 | 0.638 | 0.497 * | / | / |
| | | (0.366) | (0.180) | (0.053) | (0.058) | (0.058) | (0.099) | / | / | / |

**Note**: The standard error of the corresponding parameter is shown in parentheses; * Significant at the 95% confidence level. **Abbreviations**: *n*, number of individuals; *Na*, number of alleles; *Ne*, effective number of alleles; *Ho*, observed heterozygosity; *He*, expected heterozygosity; *uHe*, unbiased expected heterozygosity; *I*, Shannon's index of diversity; *Fis*, coefficient of local inbreeding; *AR*, allele abundance; *PAR*, private allele abundance.

### 3.2. Bottlenecks and Spatial Genetic Structure

We next conducted bottleneck analysis to determine whether these two populations have undergone population bottlenecks. We detected a significant excess of heterozygotes in the seedlings of *R. apiculata* under the IAA model according to both the sign test and the one-tailed Wilcoxon sign-rank test, suggesting that the seedlings of this population may have experienced a reduction in population size (Table 3). Our results indicated that the total population (adults + seedlings) of *R. apiculata* has experienced a bottleneck under the IAA model (Table 3).

**Table 3.** The probability of population bottlenecks of *R. apiculata* and *A. marina* under three models.

| | Model | *R. apiculata* | | | *A. marina* | | |
|---|---|---|---|---|---|---|---|
| | | Adults | Seedlings | Total | Adults | Seedlings | Total |
| Sign test | TPM | 0.450 | 0.200 | 0.430 | 0.531 | 0.142 | 0.534 |
| | IAA | 0.154 | **0.042** | 0.145 | 0.130 | 0.096 | 0.138 |
| | SMM | 0.086 | 0.269 | 0.083 | 0.225 | 0.593 | 0.069 |
| | mean | 0.230 | 0.170 | 0.219 | 0.295 | 0.277 | 0.247 |
| One-tailed Wilcoxon test | TPM | 0.422 | 0.125 | 0.422 | 0.527 | 0.230 | 0.578 |
| | IAA | 0.098 | **0.020** | **0.027** | 0.098 | 0.125 | 0.125 |
| | SMM | 0.963 | 0.473 | 0.902 | 0.973 | 0.527 | 0.980 |
| | mean | 0.494 | 0.206 | 0.451 | 0.533 | 0.294 | 0.561 |

**Note**: The one-tailed Wilcoxon sign-rank test was only used to determine heterozygosity excess. Values indicating significant bottlenecks are in bold ($p < 0.05$). **Abbreviations**: **TPM**, two-phase model; **IAA**, infinite allele model; **SMM**, stepwise mutation model.

The spatial autocorrelation analysis indicated significant spatial genetic structure at short distance classes for both seedlings and adults in the *R. apiculata* population (2–4, 4–8, 8–16, and 16–32 m; Figure 3a). Significant positive *Fij* values were observed at short distance classes among *R. apiculata* adults (2–4, 4–8, 8–16, and 16–32 m; Figure 3b). *Fij* values decreased with distance in the *R. apiculata* population. Significant positive *Fij* values were also observed for both seedlings and adults of *A. marina* (until 32 m; Figure 3c), and a similar pattern was observed for *A. marina* adults. However, *Fij* values among adults were lower than those among the total *A. marina* population, and correlations between *Fij* values and distance class were weak (Figure 3d). *Sp* statistics for each analysis are shown in Figure 2 and reveal a strong spatial genetic structure for these two species at short distances.

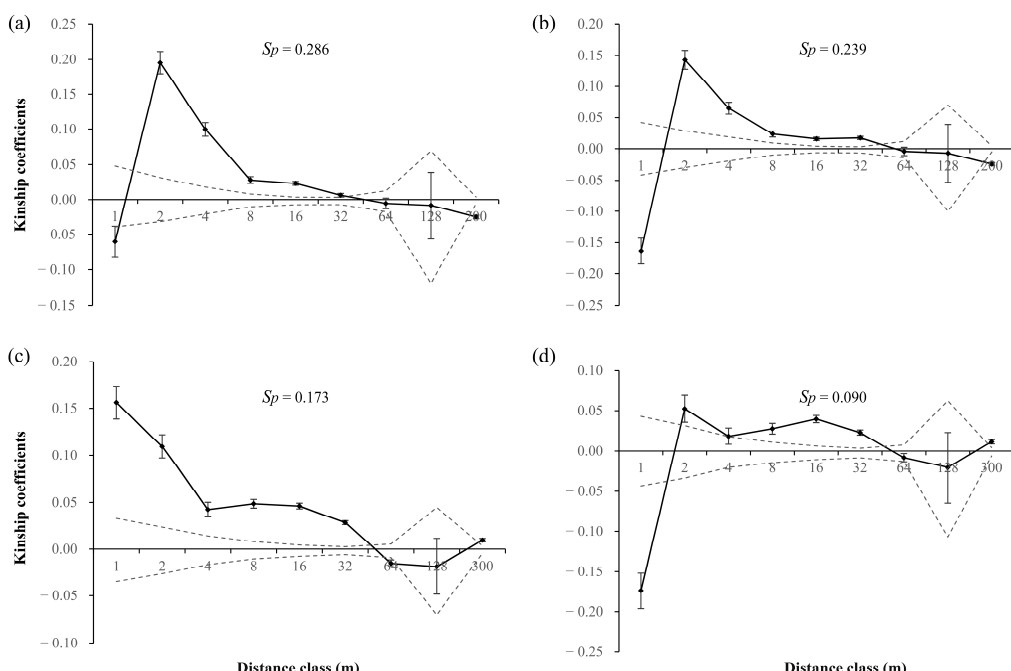

**Figure 3.** Kinship coefficients (*Fij*) in eight distance classes for all individuals (**a**) and adults (**b**) of the *R. apiculata* population and all individuals (**c**) and adults (**d**) of the *A. marina* population. The dashed lines indicate 95% confidence limits.

### 3.3. Mating System Parameters

The *tm*, *ts*, and *tm−ts* for *R. apiculata* at the population level were 1.135 (0.029), 1.111 (0.051), and 0.024 (0.042), respectively. Bootstrap analysis suggested that both *tm* (95% CI = 1.007–1.193) and *ts* (95% CI = 1.009–1.213) were greater than 1, and *tm−ts* (95% CI = −0.06–0.108) did not differ significantly from 0. This suggested the presence of high outcrossing and a relatively low proportion of biparental inbreeding in *R. apiculata*. *tm* (1.001–1.116) and *ts* (1.082–1.551) were also greater than 1 for various mother trees of *R. apiculata* (Table 4). For *A. marina*, *tm* (95% CI = 0.448–0.980) and *ts* (95% CI = 0.318–0.874) were lower than 1 at the population level, which indicated that selfing has occurred in this population. *tm − ts* (95% CI = 0.020–0.216) significantly differed from 0. This suggested the presence of significant biparental inbreeding. There was a high degree of variation in the mating system parameters of each mother tree in the *A. marina* population (Table 4).

**Table 4.** Mating system parameters for each mother tree of *R. apiculata* and *A. marina*.

| Mother Tree | Propagules | *tm* (SE) | *ts* (SE) | *tm−ts* |
|---|---|---|---|---|
| *R. apiculata* | | | | |
| Z12 | 6 | 1.001 (0.001) | 1.138 (0.053) | −0.137 |
| Z85 | 11 | 1.116 (0.041) | 1.551 (0.165) | −0.435 |
| Z110 | 6 | 1.004 (0.001) | 1.082 (0.112) | −0.078 |
| *A. marina* | | | | |
| A1 | 17 | 0.822 (0.209) | 1.027 (0.203) | −0.205 |
| A2 | 10 | 0.186 (0.151) | 0.094 (0.073) | 0.092 |
| A21 | 9 | 0.755 (0.170) | 0.691 (0.199) | 0.064 |
| A48 | 9 | 1.024 (0.001) | 0.589 (0.079) | 0.435 |
| A126 | 6 | 0.849 (0.162) | 0.818 (0.228) | 0.031 |

**Abbreviations**: *tm*, multilocus outcrossing rates; *ts*, single-locus outcrossing rates; *tm−ts*, biparental inbreeding rates.

We identified pollen donors within the *R. apiculata* quadrat for 35% (8 of 23) of the propagules at the 80% confidence level. The remaining 65% of the propagules were likely derived from pollen donors outside the quadrat. All eight propagules originated from outcrossing. These results were consistent with the mating system parameters estimated using MLTR software. However, only three (6%) propagules were identified as pollen donors within the quadrat at the 80% confidence level for *A. marina*, indicating that nearly all propagules originated from outcrossing. Mother trees of these three propagules were A48 (two propagules) and A126 (one propagule), and their *tm* was higher compared with that of other trees in the *A. marina* quadrat. The above findings suggest that most of the pollen donors were located outside of the *A. marina* quadrat. Because most pollen donors were from outside the quadrat, we did not calculate the pollen dispersal distance.

## 4. Discussion

Protections for mangroves have strengthened in China over the past two decades, and many mangrove reserves have been established [16–18]. Although much conservation effort has been devoted to increasing the area of mangrove forests, the genetic health of mangroves has been largely ignored by comparison. Small and isolated populations are often characterized by lower genetic diversity, engage in higher levels of inbreeding, and experience bottlenecks, all of which affect their resilience against a backdrop of anthropogenic threats and pressures and climate change [9,54]. Assessing the genetic diversity and mating system of such small and isolated populations of mangroves is critically important for setting reasonable restoration goals for maintaining and improving levels of genetic diversity of remaining mangrove populations [55–57].

We evaluated the genetic diversity of two mangrove species, *R. apiculata* and *A. marina*, in Tielu Harbor Mangrove Nature Reserve. There were no significant differences in the genetic diversity (*Ho* = 0.357, *He* = 0.472) of the *R. apiculata* population in Tielu Harbor with other *R. apiculata* populations. Yahya et al. [58] characterized the genetic variation of 15 *R. apiculata* populations in the Greater Sunda Islands of Indonesia using five microsatellite loci and found that *Ho* and *He* values were 0.338 (0.117–0.457) and 0.378 (0.123–0.482), respectively. The genetic diversity of *R. apiculata* in Malaysia was found to be particularly low, as the *Ho* and *He* values were 0.299 (0.194–0.483) and 0.325 (0.247–0.503), respectively [59]. Although the genetic diversity of *R. apiculata* in our study population was not low, we found that the *AR* and *PAR* values of seedlings were lower than those of adults (Table 2). Given that it takes nearly three years for propagules of *R. apiculata* to develop from flower bud primordia to maturity [60], the low genetic diversity of these seedlings may reflect the impact of recent anthropogenic activities on the population.

The genetic diversity of the *A. marina* population in our study (*Ho* = 0.195, *He* = 0.384) was lower compared with that of other populations examined in a previous study [61]. Maguire et al. [58] detected high levels of genetic diversity of *A. marina* in 14 populations across its global range using three microsatellite loci. Nearly all of the populations examined in Maguire et al. [61] had higher *Ho* values compared with the *A. marina* population in Tielu Harbor; the one exception was a Japanese population (*Ho* = 0.000). Hou et al. (unpublished data) investigated the genetic diversity of an *A. marina* population along the Sanya River, which is located close to Tielu Harbor, and the *Ho* and *He* values of this population were 0.532 and 0.575, respectively. Our data indicate that habitat degradation might have had a significant effect on the genetic diversity of the *A. marina* population in Tielu Harbor. Similarly, Salas-Leiva et al. [55] reported that the low genetic diversity (*Ho* = 0.277) of another *Avicennia* species (*A. germinans*) along the Colombian Pacific coast was associated with habitat degradation and fragmentation processes. However, we found that the *AR* and *PAR* of the seedlings of *A. marina* were higher than that of adults. The propagules of *A. marina* are small and light and can spread by sea currents [62]; thus, some of these seedlings might have originated from outside the quadrat, such as populations near Tielu Harbor (e.g., the Sanya River population and Qingmei Harbor population [39]).

We found that *A. marina* populations in Tielu Harbor had significant positive *Fis* values. High levels of inbreeding have been documented in several mangrove species, such as *Rhizophora mangle* [63], *Rhizophora stylosa* [43], *A. marina* [61], and *Kandelia candel* [64]. The high level of inbreeding observed in *A. marina* can be explained by the habitat degradation and fragmentation [55]. Considering that the population was not detected to have experienced bottleneck (Table 3) and results of the mating system and paternity analyses, we speculate that the significant positive *Fis* values and low genetic diversity in *A. marina* reflect mating between close relatives rather than a reduction in effective population size. The results of the genetic bottleneck analysis indicate that the *R. apiculata* population in Tielu Harbor may have experienced a reduction in population size induced by habitat fragmentation. Although previous studies have shown that habitat fragmentation can have significant deleterious effects on the genetic health and mating system of wind-pollinated trees [65], the outcrossing rates observed for *R. apiculata* in our study were higher than those reported for other mangrove trees [51,66]. The continuous distribution of *R. apiculata* trees in Tielu Harbor might facilitate the exchange of pollen among plants and explain the high outcrossing rate [67].

The effect of seedlings was greater on the spatial genetic structure of *A. marina* than that of *R. apiculata* (Figure 3). Pronounced spatial genetic structure has been detected over short distances within populations of mangrove species given that many propagules become established around the mother tree [51]. The pollen and propagule dispersal of *R. apiculata*, a viviparous species, is limited; similar observations have been made for another viviparous mangrove plant, *K. candel* [51,68]. However, because each *R. apiculata* individual occupied a large space, virtually no seedlings can grow further up within 2 m of the mother tree. As discussed above, this creates significant genetic structure over short distances, as was observed in adults of *R. apiculata* (2–32 m, Figure 3b). The spatial genetic structure of *A. marina* within 2 m changed significantly depending on whether seedlings were included in the analysis. This can be explained by the amount of space occupied by adult trees. Adult *A. marina* trees are small, and seedlings can grow around their mother trees. Thus, when seedlings were excluded from the analysis, genetic structure was not significant within 2 m, but when they were included, genetic structure was significant.

We studied the genetic diversity, spatial genetic structure, and mating system of two mangrove species, *R. apiculata* and *A. marina*, in a heavily disturbed area in Tielu Harbor, China, using nuclear microsatellite markers. Microsatellite markers are widely used in population genetic studies of mangrove plants, such as *R. apiculata* [58,59], *K. candel* [51,68], *A. germinans* [55–57] and *A. marina* [61], to estimate genetic diversity, genetic structure and mating system. This indicates the usefulness of microsatellites in the genetic analysis of mangrove plants. The findings from our studies and field investigations suggest that these two mangrove species require distinct restoration and rehabilitation approaches. Given that the diversity of the *R. apiculata* population in Tielu Harbor is similar to that of other populations [58,59], the propagules produced by local adult trees could be used for mangrove restoration. However, for this approach to succeed, the connectivity among individuals needs to be maintained, and this includes the connectivity between newly restored mangroves with existing mangroves, to minimize the effects of inbreeding on future generations. Connectivity among individuals is also considered important in the in-situ conservation of another mangrove plant, *A. germinans* [57]. The low genetic diversity and significant inbreeding of *A. marina* in Tielu Harbor suggest that the genetic health of the *A. marina* population is poorer than that of the *R. apiculata* population [61]. Consequently, the use of propagules from the *A. marina* population in Tielu Harbor for mangrove restoration is likely insufficient for ensuring the long-term persistence of restored *A. marina* populations. In a previous study, Salas-Leiva et al. [56] considered that reforestation using propagules from different populations would improve the maintenance of genetic diversity and the viability of the reforested population in the short and medium term. Propagules from numerous populations with high genetic diversity are needed to ensure that restored

*A. marina* populations are resilient enough to persist in the face of anthropogenically driven habitat degradation and climate change [28,69].

Mangrove conservation and restoration efforts in China have generally been successful over the last several decades. Approximately 67% of mangroves are now within nature reserves, and 43 mangrove protected areas have been established [16]. Studies of the genetic diversity, fine-scale genetic structure, and mating system of mangroves are needed to ensure the success of mangrove conservation efforts. In this study, we only selected limited seedlings and propagules to minimize the impact on the capacity for population renewal. We believed that if more seedling and propagule samples can be taken, the results should be more reliable. Furthermore, additional studies are needed to evaluate the genetic diversity of populations following mangrove restoration, as well as monitor the status of populations. Such studies of genetic diversity and structure would provide valuable insight into whether restoration efforts are having their intended effect of increasing the resilience of restored areas.

**Supplementary Materials:** The following are available online at https://www.mdpi.com/article/10.3390/d14020115/s1, Table S1: Information of 18 nSSR primer pairs, Table S2: Summary of the AMOVA results for adults and seedlings in *R. apiculata* population, Table S3: Summary of the AMOVA results for adults and seedlings in *A. marina* population.

**Author Contributions:** Conceptualization, W.L. and S.Y.; methodology, W.L. and S.Y.; validation, W.L. and Z.Z.; formal analysis, W.L.; investigation, W.L. and S.Y.; data curation, W.L. and Z.Z.; writing—original draft preparation, W.L.; writing—review and editing, W.L., X.H. and S.Y.; visualization, W.L.; supervision, S.Y.; project administration, S.Y.; funding acquisition, S.Y. All authors have read and agreed to the published version of the manuscript.

**Funding:** This research was funded by the National Key Research and Development Program of China (2017YFC0506103); Ocean Public Fund Research Project Grants of State Oceanic Administration (No. 201305021-2); and National Natural Science Foundation of China (No. 30972334).

**Institutional Review Board Statement:** Not applicable.

**Data Availability Statement:** Not applicable.

**Acknowledgments:** We thank editors and three anonymous reviewers for their suggestions.

**Conflicts of Interest:** The authors declare no conflict of interest.

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
