# Peer review of "Genetic Diversity and Mating System of Two Mangrove Species (Rhizophora apiculata and Avicennia marina) in a Heavily Disturbed Area of China"

_diversity, doi:10.3390/d14020115_

Round 1

Reviewer 1 Report

The manuscript examines genetic diversity parameters of two mangrove species in Tielu Harbor, Sanyo City China, with the aim to determine the genetic health of plants in sampled quadrats. Since the sampling is restricted to a single sampling quadrat per species, the data is limited to this sampling location and has little broader scientific significance. For such a local study, a better sampling scheme should have been used. For example, numbers of progeny per array and numbers of mother trees should be considerably greater to give good unbiased estimates of mating system analyses. I have listed below comments on the manuscript.

Lines 115-116. Morphological traits should be eliminated as these are not discussed further in the manuscript.

Line 131 More detail on propagule tissue used for DNA extraction needed to avoid sampling maternal tissue.

Lines 147-148 Why compute Fis with FSTAT and then compute the 95% confidence limits with GDA? Either use FSTAT, or GDA for both the estimate and its significance. FSTAT will give a probability that Fis differs from zero.

Lines 158-159 How were distance classes chosen; set distances, or equal representation in each distance class? This should be stated in the text, with numbers of pairwise genetic distances in each distance class

Line 161 Explain why the Sp statistic is computed.

Table 1 and preceding text. Note that this is for combined adult and seedling plants

Table 2 APR should be corrected to PAR

Lines190-197 Several diversity parameters are described as being different between seedlings and adults. Are these differenced significant? Without standard errors, this cannot be judged.

Line 206-207. Th authors should be clear that Bottleneck is not detecting a significant excess of heterozygotes in seedlings of R. apiculata. This would not correspond with the heterozygote deficiency in Table 2.

Line 226. How does spatial structure exist at such short distances (1-2m) if R apiculata trees are large and seedlings do not grow up under the canopy of adults see lines 328-329: “However, because each R. apiculata individual occupied a large space, virtually no seedlings survived within 2 m of the mother tree”?

Fig 3 Where are standard error bars for each class kinship coefficient?

Lines 24-235 Explain how outcrossing rates can be greater than 1

Lines 239-240 Please show the population level outcrossing rates to judge whether selfing is likely to have occurred rather than just biparental inbreeding.

Numbers of propagules per tree are mostly low in these analyses and likely do not give robust estimates of outcrossing rates. See Ritland for guidelines on numbers of progeny per array.

Lines 251-252 This estimate of pollen flow distance is highly biased. As 65% of pollen parents came from outside of the stand, it is impossible to know the pollen distance for these.

Lines 279-280 Table 2 does not provide confidence limits around the estimates of AR and PAR, so we cannot judge whether seedling values were lower than adult values, so the following sentence is speculation.

Lines 295-297 Please back this up with objective support. How do you claim debris is affecting A. marina growth performance?

Lines 298-304 These are not alternatives. It is sufficient to say that some of the seedlings could be drift from outside the quadrat.

Lines 305-306 Adults of R. apiculate did not have a significant Fis, so it should not be invoked here.

Line 311 Again Fis was not significant, so you cannot talk about high inbreeding for R apiculata adults.

Lines 314-319 It is difficult to follow the argument here. On the one hand the authors argue for high inbreeding coefficients and on the other hand exceptionally high outcrossing rates for R. apiculata?

Line How do you argue that small effective population size is not important in determining the high Fis in A marina?

Lines 334-336 How does this relate to the Sp index that showed spatial structure at 1-2m (see my earlier comment)?

Lines 346-347 This is not justified by the data (see my earlier comment on bias of pollen dispersal distances).

Author Response

Dear Reviewer 1,

Thanks very much for taking your time to review this manuscript. We are very grateful for your comments on the manuscript. According to your advice, we amended the relevant part of the manuscript. Please see the attachment.

Best Regards.

sincerely yours,

Wen-Xun Lu

Reviewer 2 Report

I congratulate the authors for this interesting contribution. The article is clear, well organized along the sections and provide results that could be of great interest for researchers related to mangroves and wetlands research. I  just suggest to include a comparison with similar studies for other species and regions, e.g. see Millán-Aguilar et al. (2016), Forests, 7(197) and Salas-Leiva et al. (2009), Aquat. Bot, 91, related to Avicennia germinans located in perturbed and preserved sites.

Author Response

Dear Reviewer 2,

Thanks very much for taking your time to review this manuscript. We really appreciate all your generous comments and suggestions! Please see the attachment.

Best Regards.

sincerely yours,

Wen-Xun Lu

Reviewer 3 Report

Although genetic diversity from mangrove plants have been studied since 2000s. This study is an important, well researched and well written article. The authors are to be complimented.

I have noted major revision to catch your attention.

I suggest you to measure the genetic structure by analysis of molecular variance (AMOVA) based on the F-statistics. You will obtain the genetic diversity between populations each species, within individuals in the populations, and among individuals within populations.

Other comments:

  1. Legend Figure 1 c, I could not see the propagules of R. apiculata, they are flowers of R. apiculata. Please revise or change the photo of R. apiculata propagule
  2. Please discuss the implication of your study to conservation of both species and how the restoration efforts influence the genetic diversity. Since we have noted that the first world record of mangrove restoration efforts dates back to the 1950s in China.
  3. You sampled the small number of seedlings compared to adult trees, I think you provide the rationale of this issue and readers understand your results.
  4. Please discuss briefly microsatellite markers evaluate the mating system of both species and the possibility to adapt to other mangrove species.

Author Response

Dear Reviewer 3,

We are very grateful for your comments on the manuscript. According to your advice, we amended the relevant part of the manuscript. All of your questions were answered one by one. Please see the attachment.

Best Regards.

sincerely yours,

Wen-Xun Lu

Round 2

Reviewer 1 Report

Accept as is, but please note my previous comments on experimental design

Author Response

Dear Reviewer 1,

First of all, thank you for your kind decision; Secondly, thank you for your constructive suggestions on the manuscript, which makes the manuscript perfect and close to publication. As you pointed out, the rational design of the experiment is extremely important, which also provides ideas for us to carry out relevant research in the future.

Thank you very much!

Sincerely yours,

Sheng-Chang Yang

Reviewer 3 Report

The authors have addressed all my comments and suggestions, I now recommend to accept the revised manuscript.

Author Response

Dear Reviewer 3,

Firstly, thank you for your kind decision; Secondly, thank you for your constructive suggestions on the manuscript, which makes the manuscript perfect and close to publication. According to your suggestion, we improved the revised draft. Thank you very much!

Sincerely yours,

Sheng-Chang Yang